

# Potato bacterial wilt in Ethiopia: history, current status, and future perspectives

Gebrehanna Lemma Tessema and Hussen Ebrahim Seid

Horticulture Research Department, Holetta Research Centre, Ethiopian Institute of Agricultural Research, Addis Ababa, Ethiopia

## ABSTRACT

**Background:** Potato is an essential food staple and a critical tuber crop for rural livelihoods in Ethiopia, where many pathogenic pests are threatening production. Bacterial wilt, also known as brown rot of potato, ranks among the diseases that most affect many potato farmers in Ethiopia and the disease losses dramatically threatening the vibrant potato sector even in the highlands of the country where it has been uncommon so far.

**Methodology:** To devise a strategy towards boosting potato productivity in Ethiopia where food insecurity is most prevalent, production constraints should be investigated and properly addressed. Hence, we have used existing reviews and reports on the subjects, such as textbooks, and proceeding and conference abstracts in Plant Protection Society of Ethiopia; Web of Science; Google Scholar; Research Gate and CIP's database to document most relevant information on the occurrence, distribution, and disease management of bacterial wilt in Ethiopia.

**Results:** Provision of comprehensive information on potato bacterial wilt occurrence, distribution, and management techniques are crucial for potato growers, researchers and stakeholders engaged on potato industry. In this review, we provided insights on the history, status, and future perspectives of potato bacterial wilt in Ethiopia.

**Conclusions:** Awareness of potato bacterial wilt and integrated disease management approaches could bring a fundamental impact to the farming community mostly to smallholder farmers in developing countries. This document compiled such imperative information targeting bacterial wilt management techniques to ensure food security.

Corresponding author
Gebrehanna Lemma Tessema,
lematessema@gmail.com

## INTRODUCTION

Understanding and managing bacterial wilt that devastates potato, a major food staple crops worldwide, is of a great importance. This is because of the need for food security to the ever-growing world population especially in low-income countries where climate change, disease emergence and expansion, and the increasing threat of alliance pests and disease are challenging the food security. Potato bacterial wilt, the second most important pathogenic plant bacteria has been a threat for tropical agriculture due to its wide host range, worldwide distribution, and limited disease control prudence (*Mansfield et al., 2012*; *Champoiseau, Jones & Allen, 2009*; *Su et al., 2021*).

Among many other crop production constraints, plant pests and diseases pose a threat to food security because they can damage crops, thus reducing the availability and access to food, increasing the cost of production (*FAO, 2017*). The problem is worsening more than ever before due to increasing global trade and climate change that resulted these pests moving from their native environments to the newly climate favored environment (*Caruso et al., 2005*). This is because climate circumstances put forth a significant influence over the spreading, life cycle, duration, infestation pressure, and the overall occurrence of major agricultural pests and diseases (*Kocmankova et al., 2009*). Furthermore, plant pests and diseases are responsible for losses of 20% to 40% of global food production (*FAO, 2017*).

Bacterial wilt also known as brown rot of potato, is the most devastating disease of many economically important crops worldwide (*Elphinston, 2005*; *Swanson et al., 2005*; *Champoiseau et al., 2010*). It is caused by the *Ralstonia solanacearum* (Smith) species complex, that is a soil-borne pathogen notorious for its virulence, wide host range, and broader geographical distribution (*Fegan & Prior, 2005*; *Abdurahman et al., 2019*). Moreover, once the pathogen successfully establish to potato fields, eradication of the bacterium is very difficult, because of its long persistence nature in water, deep soil layer rhizosphere and plant roots, plant debris, as well as volunteer potato and other host plants for long period before coming in to contact with a new host (*Graham, Jones & Lloyd, 1979*; *Graham & Lloyd, 1979*; *Priou et al., 1999*; *Mihovilovich et al., 2017*). Their success as agricultural pests can be attributed to their adaptability to variable environments and their ability to survive adverse environmental conditions for extended periods of time (*Caruso et al., 2005*; *Milling et al., 2009*).

Though potato bacterial wilt is a soil-borne disease, the spread and prevalence as well as disease outbreak could be aggravated by many biotic and abiotic factors such as poor agricultural practices, unlimited animal and human movement to farming fields, lack of knowledge on appropriate field sanitation procedures, planting pathogen infected seed year after year, weak seed quarantine rules, soil acidity, and environmental variations (*Gorfu, Woldegiorgis & Kassa, 2013*; *Tafesse et al., 2021*; *Tessema et al., 2020*; *2022*). It was evidenced that environmental temperature also has direct effect on the success of pathogen invasion to varying host species (*Wei et al., 2015*).

In Ethiopia, the disease threatens the vibrant potato sector even in the highlands of the country, where the disease was uncommon and seed is being sourced out to different parts of the country (*Abdurahman et al., 2017*; *Tessema et al., 2022*). The lack of reliable seed source and the heavy reliance on uncertified tubers as planting material resulted frequent re-infection of healthy fields. Subsequently, such infected fields re-infect the crops whether we have planted healthy or infected plants. This review therefore provides comprehensive coverage of currently available international research data that illustrates the occurrence, distribution, and pathogenic effect of bacterial wilt. Disease management tactics relevant to manage the pathogen are also emphasized. The review also intended to summarize and document the history, status, and future perspectives of potato bacterial wilt in Ethiopia useful to potato growers, researchers and other stakeholders engaged in potato industry.

## SURVEY METHODOLOGY

To survey potato bacterial wilt occurrence, distribution, importance, and perspectives for future agriculture, we have searched the following databases: existing reviews and reports on subject, textbooks, proceeding and conference abstracts in Plant Protection Society of Ethiopia, Web of Science, Google Scholar, Research Gate, CIP's database on potato diseases, annual and progress reports of the Ethiopian Institute of Agricultural Research, and Holetta Agricultural Research Centre. We have used the following search terms: Bacterial wilt, *Ralstonia*, brown rot, ELISA, detection methods, biological control, integrated disease management, bacterial wilt in Ethiopia, disease prevalence, disease incidence, seed health, climate change, epidemiology, disease symptom, crop loss, race, biovar, RSSC, characterization, diagnosis, and distribution. We also used references cited by the articles obtained by these methods to search for relevant additional material in the subject matter.

## OCCURRENCE AND DISTRIBUTION OF *R. SOLANACEARUM*

Historically, bacterial wilt occurrence was first reported in Ethiopia in 1956 from Kaffa province (Kaffa administrative zone) (*Stewart, 1956*). Since then, it was reported from different parts of southern, west, southwest, northern, and central highlands of the country (Table 1; Fig. 1) with more emphasis in central high lands where major seed potato is being distributed to the country (*Gorfu, Woldegiorgis & Kassa, 2013*; *Abdurahman et al., 2017*; *Sharma et al., 2018*, *Tessema et al., 2020*; *2022*). The disease transmission was slow and not a plague too many potato farms at a time, so that no one gave due attention to the pathogen and by now it becomes a serious production threat for the farming community (*Gorfu, Woldegiorgis & Kassa, 2013*; *Sharma et al., 2018*; *Tafesse et al., 2021*). The spread of the disease to different parts of the country was traced from latently infected seed tubers and currently the pathogen is serious constraint to more than 5 million potato farmers of Ethiopia (*CSA, 2018*; *Abdurahman et al., 2017*). In addition to the movement of latently infected seed potato from place to place, there is scarcely well organized and responsible body that implements the biosecurity check list (people, vehicles, and agricultural inputs) that carry pathogens, insects and weed seeds on to and around the farm that infect previously clean fields (*PHAU, 2018*). Bacterial wilt monitoring system by itself has various challenges and it depends on visual observation (*Tafesse et al., 2020*).

Afterwards, different scholars made tremendous efforts to study the spread, prevalence and importance of bacterial wilt to the Ethiopian farming community from simple disease identification to advanced molecular level (Table 1). However, it is not an easy task to study every aspect of the disease due to the complex nature of *R. solanacearum* species complex (*Hayward, 1991*; *Safni, Subandiyah & Fegan, 2018*; *Fegan & Prior, 2005*; *EPPO, 2018*). The study by *Abdurahman et al., 2017* on phylogenetic analysis from potato isolates in Ethiopia using multiplex PCR and phylogenetic analysis of partial endoglucanase gene sequences identified all the isolates as phylotype IIB sequevar 1 strains. Similarly, this strain (PIIB-1) caused clonal brown rot epidemics *via* latently infected seed potato in Peru

**Table 1 Prevalence and distribution of potato bacterial wilt to different parts of Ethiopia.**

| Locality (Zone) | Host crop | Disease prevalence (%) | Year reported | Reference |
|---|---|---|---|---|
| Agew Awi | Potato | 25 | 2008 | *Bekele et al. (2011)* |
| East Arsi | Potato | 0.8–91.6 | 1981; 1985; 1993 2020 | *SPL (1981)*, *Kassa & Hiskias (1994)*, *Abebe (1999)*, *Tessema et al. (2020)* |
| East Shoa | Potato | 0.8–63 | 1985–87; 1987; 1994 | SPL (1987), *Kassa & Hiskias (1994)* |
| Gamo gofa | Potato | 97; 10–80 | 2015/16 | *Abdurahman et al. (2017)* |
| Guraghe | Potato | 0-23.12 | 2015/16 | *Tessema et al. (2020)* |
| Hadiya | Potato | 0–33 11-89 | 2015/16 | *Tessema et al. (2020)* |
| Holetta/ Welmera | Potato | 11; 63; 10.87–90 | 1994; 1996; 2020 | *Kassa & Hiskias (1994)*, *Tessema et al. (2020)* |
| Jimma | Potato | | 2007 | *Kurabachew, Assefa & Hiskias (2007)* |
| Kaffa | Potato | 23–50 21–78 | 1956; 1967; 2012–14 | *Stewart (1956)*, *Stewart & Yirgou (1967)* |
| Kembata Tembaro | Potato | 42.5 | 2015/16, 2018 | *Tessema et al. (2020)*, *Tafesse et al. (2018)* |
| North Gonder | | 50–100 | 2008 | *Bekele et al. (2011)* |
| North Shoa | Potato | | 1967; | *Stewart & Yirgou (1967)*, *Tessema et al. (2020)* |
| Sidama | Potato | 62.5 | 1996/97; 2007 | *Abebe (1999)*, *Kurabachew, Assefa & Hiskias (2007)* |
| Silte | Potato | | 2015/16 | *Tessema et al. (2020)* |
| South Gonder | Potato | 20 | 2008 | *Bekele et al. (2011)* |
| South west Shoa | Potato | 60 | 2015/16 | *Tessema et al. (2020)* |
| West Arsi | Potato | 25–75 | 1996/97; 2011–2014; 2015/16 | *Abebe (1999)*, *Kassa (2016)*, *Kassa & Chindi (2013)*, *Tessema et al. (2020)* |
| West Gojam | Potato | 66.7–100 | 2008 | *Bekele et al. (2011)* |
| West Shoa | Potato | 1.5–82.5 | 1985–1987; 2007; 2015 | *Kassa & Hiskias (1994)*, *Kurabachew, Assefa & Hiskias (2007)*, *Tessema et al. (2020)* |
| Wolayta | Potato | – | 2007 | *Kurabachew, Assefa & Hiskias (2007)* |

(*Gutarra et al., 2017*). On the other hand, strains of *R. solanacearum* highlighted that there was epidemiological links between southwest Indian Ocean and Africa, Americas or Europe (*Yahiaoui et al., 2017*), thus indicating how the strains of *R. solanacearum* are complex and distributed across continents.

Based on the phylotyping scheme, which is a new scheme for classifying *R. solanacearum* as of *Fegan & Prior (2005)* members of the *R. solanacearum* species complex was categorized to four subdivisions of phylotypes corresponding to the four genetic groups identified *via* sequence analysis. Among these four major phylotypes, phylotype I, II and III were reported from Ethiopia and phylotype III contains primarily isolates from Africa and surrounding islands, strains belong to biovars 1 and 2T (*Abdurahman et al., 2017*). On the other study, *Lemessa, Zeller & Negeri (2010)* reported

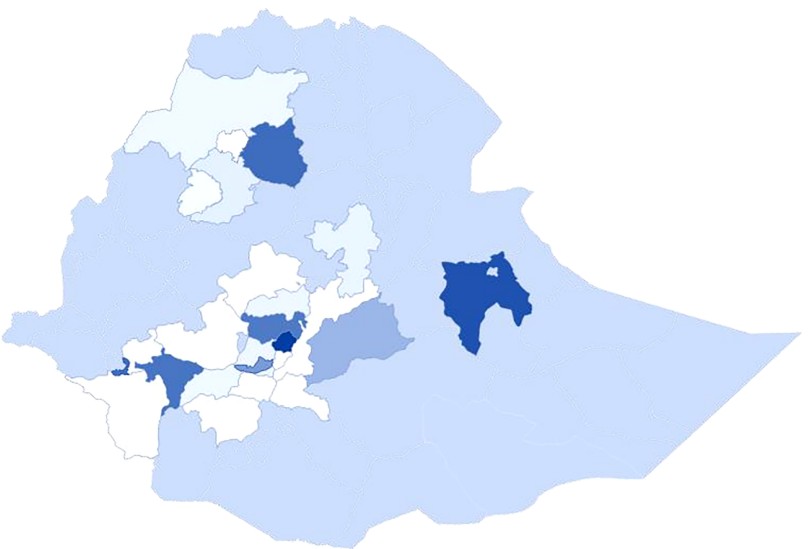

**Figure 1 Bacterial wilt distribution in different administrative zones of Ethiopia.**

**Table 2 Strains of *R. solanacearum* reported from Ethiopia assessed by PCR.**

| Geographical location | Original host | Phylotype/ Biovar | Race | Reference |
|---|---|---|---|---|
| Chencha, Holetta, Jeldu, Haramaya, Shashemmene | Potato | IIB | 1 | *Abdurahman et al. (2017)* |
| Jimma, Holetta, Ginchi, Jeldu, Shashemene, Awassa | Potato | I | 1 | *Lemessa, Zeller & Negeri (2010)* |
| Bako, Jimma, Qarsa, Kombolcha, Agaro, Ambo, Shashemen, Gedo, Guder, Awassa, Holetta, Ginchi | Potato | II | 3 | *Lemessa, Zeller & Negeri (2010)* |

Ethiopian biovar 2 strains may fall in phylotype II and be American origin while most of the biovar 1 strains fall in to phylotype III and be African origin (Table 2).

*R. solanacearum* is a widespread pathogen to tropical, subtropical, and warm temperate areas throughout the world, though its occurrence has been reported from temperate zones (*EPPO, 2020*).

The distribution continues with various means to different countries and more than 35 African countries are currently infected with *R. solanacearum* (*EPPO, 2020*; *Abdurahman et al., 2019*). In 1999, the bacterium race 3 biovar 2 was imported through infected *Pelargonium zonale* cuttings to United Kingdom and from September to December 2000, bacterial wilt was identified in several *Pelargonium* nurseries in Belgium, Germany, and the Netherlands. The disease spread was serious due to its limited symptomatic clues while visualizing infected plants (*Champoiseau, Jones & Allen, 2009*). On the other hand, the increased globalization of crops and processing industries also promotes the side-effects of more rapid and efficient spread of plant pathogens threatening the worlds' food security as well (*Lenarcic et al., 2014*).

Furthermore, many developing countries like Ethiopia do not have well-documented seed system policies whereas other have the legal framework but have not yet implemented the policies (*Schulz et al., 2013*). Studies signified that despite a regulatory regime that imposes strict rules on the production and trade of planting materials for vegetatively propagated crops, the market is largely unregulated because of weak enforcement capacity. Instead, producers of vegetatively propagated crops planting materials signal quality to farmers through trust, reputation, and long-term relationships (*Gatto et al., 2021*).

Lack of such seed legislation rules resulted the fastest spread of the disease with no territory limits (*Lambert, 2002*). For example, phylotype IIB sequevar 1 (PIIB-1), formerly referred to as R3b2 and known as the causal agent of potato brown rot was distributed from latently infected seed potato from south America to other world (*EPPO, 2018*; *Charkowski et al., 2020*). Likewise, *R. solanacearum* R3bv2 was imported in the United States and Europe unintentionally in geranium cuttings (*Janse et al., 2004*; *Williamson, Hudelson & Allen, 2002*; *Swanson et al., 2005*) and the problem was severed in U.S. after 2003 when geranium plants imported from Kenya, Costa Rica and Guatemala.

The causative agent of the outbreak, R3bv2 was listed on the Agricultural Bioterrorism Protection Act of 2002 because of its potential impact on the U.S. potato industry (*Lambert, 2002*; *Williamson, Hudelson & Allen, 2002*). The spread of the pathogen in Ethiopia is highly accelerated by distributing latently infected seed tubers from region to region (*Abdurahman et al., 2017*).

The *R. solanacearum* species complex strains have dissimilar existences and diversified pathogenic behavior like surviving in heterogeneous niches and asymptomatic plants complicated the disease management efforts of scientists across the world (*Alvarez, Biosca & Lopez, 2010*; *Fegan & Prior, 2005*). *R. solanacearum*, a soil-borne pathogen that infects the plants through the root system trigger evolving anomalies in the root system (*Xue, Lozano-Duran & Macho, 2020*) and subsequently spread to the whole plant that finally results wilting of the host plant (Fig. 2).

## HOST RANGE OF BACTERIAL WILT

*R. solanacearum* species complex (RSSC) is soil-borne phytopathogenic bacteria that is known to invade more than 200 host species in more than 50 plant families worldwide (*Elphinston, 2005*; *Coupat et al., 2008*; *Mansfield et al., 2012*). For instance, tomato, potato, tobacco, and eggplant from solanaceous family, ground nut and French bean from leguminous plants; banana and ginger from mono-cotyledons plants; eucalyptus, olive, mulberry, and cassava from tree and shrub plants are some of the hosts (*Hayward, 1994*; *Ji et al., 2007*; *Champoiseau et al., 2010*; *Liu et al., 2017*; *Guji, Yetayew & Kidanu, 2019*). Thorn apple and nightshade are also other common hosts that harbor the disease (*PHAU, 2018*). Recently, the recording work for maximum limits of host plants are not ended yet and new host plants are being described by different scientists at every moment from each corner of the world (*Chandrashekara & Prasannakumar, 2010*). Besides, several new sequevars have been identified and recorded in recent studies that cause plant wilts (*Sarkar & Chaudhuri, 2016*; *Yahiaoui et al., 2017*; *Gutarra et al., 2017*).

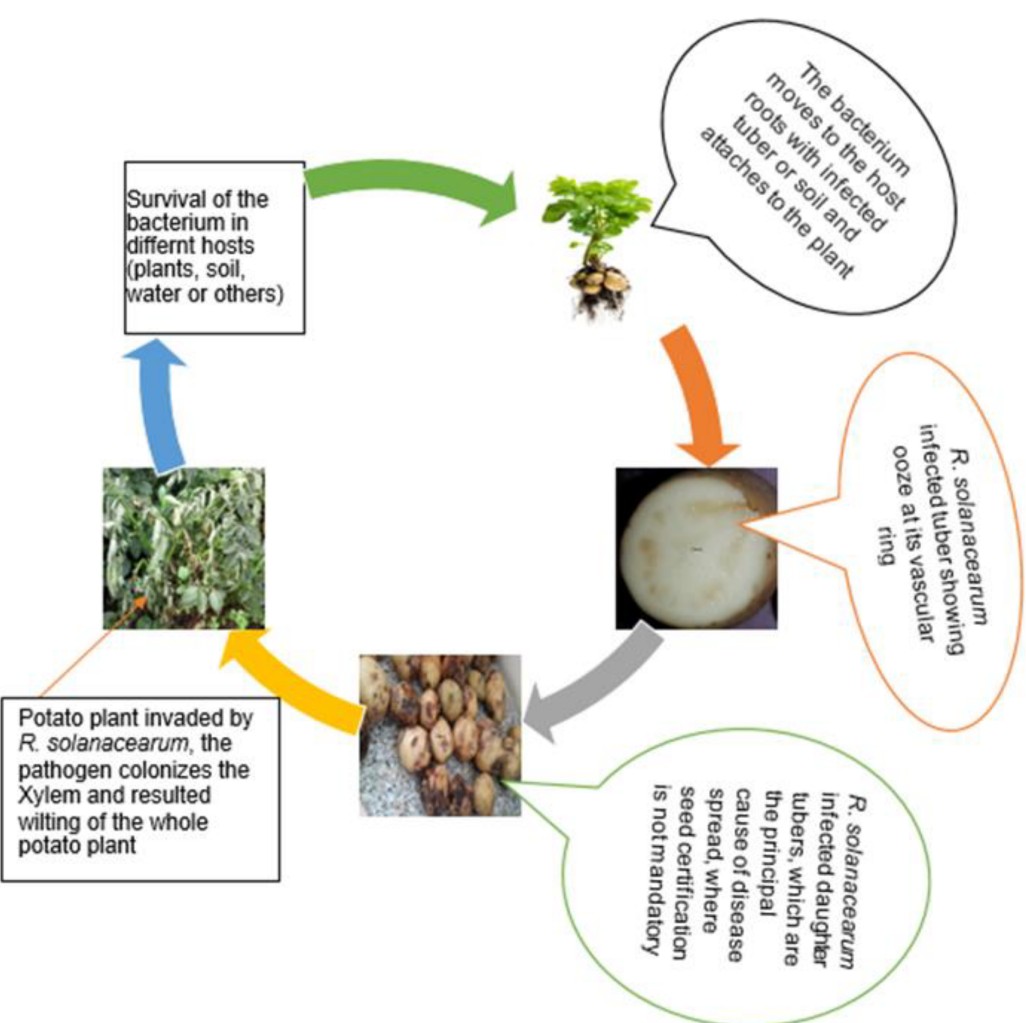

**Figure 2 A schematic diagram of *R. solanacearum* life cycle illustrated from various sources and pictures collected by authors in Ethiopia.**

## IMPORTANCE OF BACTERIAL WILT

Gram-negative bacterium, *R.* (Formerly known as *Pseudomonas*) *solanacearum* (*Yabuuchi et al., 1995*) is one of the most destructive plant pathogens (*Elphinston, 2005*; *Safni et al., 2014*; *Su et al., 2021*). *R. solanacearum* has already been a quarantine pest and added to the front line of invasive species compendium in the European and Mediterranean Plant Protection Organization (EPPO) due to its threatening power of million's livelihoods and the environment worldwide (*EPPO, 1992*). It also has long been a scourge of tropical agriculture due to its wide host range, worldwide distribution, and limited disease control providence (*Champoiseau, Jones & Allen, 2009*; *Su et al., 2021*). It is listed as the second most important plant pathogenic bacteria in 'top 10' lists of bacterial diseases in terms of economic and scientific importance ranked by the world bacteriologists with only preceded by *Pseudomonas syringae* pathovars even though priorities and importance could vary in the locality across continents and disciplines (*Mansfield et al., 2012*). Among many

plant bacterial diseases, seven bacterial diseases were categorized as the most important for potato production worldwide because of their destruction to the economic part of the crop (tuber). With these, bacterial wilt and black leg were considered as the major imperative in most potato production areas whereas potato ring rot, pink eye, and common scab considered as minor (*Charkowski et al., 2020*).

The disease incidence in some parts of Ethiopia from 1985 to 1987 report was 0.8% to 21% in Tsedey farm, Zewai and 1.5% to 24% in Bako areas. In recent years, the disease prevalence raised to above 90% in most potato growing areas of Ethiopia (*Abdurahman et al., 2017*; *Tessema et al., 2020*). Furthermore, the disease was important only in warmer areas of the country, but the occurrence of bacterial wilt in the highlands like Tsedey farm laid a base to consider and gained attention even at higher altitude potato domain areas would have probability to be infected with the pathogen (*Kassa & Hiskias, 1994*) and recent studies confirmed that Ethiopian highlands with an altitude of above 3000 m.a.s.l. are found infected with *R.solanacearum* (*Tessema et al., 2022*). Plant bacterial wilt causes a huge amount of annual (32%) crop loss in Ethiopia with its epidemic levels (*Kassa & Chindi, 2013*; *Sharma et al., 2018*). The pathogen, causes dual harm to the farming community by causing considerable yield loss and it raises the crop management cost of the farming society worldwide (*Blomme et al., 2014*). The significance of the pathogen for potato crop has many dimensions because it damages the most important and economic part of the crop and causes severe economic losses along the whole potato value chain (*Charkowski et al., 2020*). Reports signified direct yield loss of 30% to 90% in potato (*Karim & Hossain, 2018*). As a result, many countries miss their alternatives to access quality seed and import or export opportunities due to the pest quarantine law of the territories. In addition to yield loss, the bacterium affects post-harvest processing quality of tubers and thereby impeding the ultimate consumers' preference (*Dagne & Tigist, 2018*).

In the current scenario, bacterial wilt has a worldwide distribution, found in almost all continents and threatening the agricultural community of the globe (*EPPO, 2020*; *Gorfu, Woldegiorgis & Kassa, 2013*). Likewise, bacterial wilt distributed to many African countries and became one of serious potato production constraints in Burundi, Cameroon, Egypt, Ethiopia, Kenya, Madagascar, Nigeria, Rwanda and Uganda, to mention some (*Kabeil et al., 2008*; *CTA, 2014*; *Elnaggar et al., 2018*; *Abdurahman et al., 2019*; *Tessema et al., 2020*). Major seed potato exporting countries like Canada, Netherlands, France and United Kingdom (*van Loon Kees, 2007*) as well as seed potato importing countries of Africa like Algeria, Egypt, Morocco and Tunisia are highly influenced by potato bacterial wilt. The seed trade share of Africa was only 2% which might be due to seed quality issues including quarantine diseases like *R.solanacearum* that leads to decline the global trade share of the continent as well (*CTA, 2014*). In Egypt, a latently infected tuber has resulted in a strong decline of potato export to Europe (*Priou et al., 1999*) and Egyptian potato export fell from a peak value of US$ 102.12 million in 1995 to US$ 7.7 million in 2000 due to potato brown rot quarantine restriction, imposed by the European Union, that was accounted for about 70–90% of Egypt's potato exports (*Kabeil et al., 2008*).

In the coming near future of 2050, the world's population is projected to be increased by one-third and leads to additional two billion people will live in developing countries (*FAO,*

*2017*). Agriculture must therefore transform itself and agricultural production should be increased by 60% to feed these huge growing global populations (*FAO, 2013*). Increasing crop production through improved plant protection could be one aspect of the approaches to solve such food security burdens by saving pathogen induced yield losses that ranges between 20% to 40% (*PPSE, 2009*; *Savary et al., 2012*). The total global potential losses due to pests may differ among crops and varied from 50% in wheat to more than 80% in cotton (*Oerke, 2006*). Post-harvest losses account for 30% to 40% yield loss due to diseases and sub-standard quality caused by scanty post-harvest handling (*Savary et al., 2012*; *Oerke, 2006*; *Habtegebriel, 2017*). Post-harvest loss of potato along the value chain in Ethiopia was 20–25% (*Tadesse, Bakala & Mariam, 2018*). For instance, the yield loss on important food crops by Arthropod's pests only were estimated 18–20% annual production worldwide which was valued at more than US$ 470 billion (*Sharma, Kooner & Arora, 2017*). Despite the economic importance and crop losses attributed to these diseases, research and extension efforts have realized limited achievements to struggle the diseases that threaten the agricultural sector (*Kassa, 2019*).

## DISEASE CONTROL

Disease control has different principles and tactics applied before or after infection. These disease control principles are exclusion, eradication, protection, immunization, avoidance and therapy among many others that has been illustrated by different scholars (*Maloy, 2005*).

### Host resistance or using clean seed

Studies are indicating that there is no high level of resistance to bacterial wilt in potato cultivars, though some cultivars are less susceptible than others and can give high yields in the existence of the pathogen (*Groza, Bowen & Kichefski, 2004*; *Mihovilovich et al., 2017*). On the other hand, one strain is aggressive than the other and host resistance is woefully possible due to its complexity among species as well as existence of high genetic diversity of the bacterium (*Fegan & Prior, 2005*; *Swanson et al., 2007*).

In CIP breeding program for bacterial wilt resistance, moderate to high levels of resistance were reported in some potato cultivars even though high frequency of latent infection in tubers was considered as a problem and latent infection is still responsible for disease spread and overcoming of cultivar resistance (*French, Anguiz & Aley, 1998*; *Priou, Aley & Gutarra, 2005*). According to *Boschi et al. (2017)* bacterial wilt resistance enhanced in potato through expression of Arabidopsis EFR and introgression of qualitative resistance from *Solanum commersonii* would be promising strategy to bacterial wilt resistance in potato. The most auspicious approaches fall under the headline "New Genomic Technologies" (NGT) where host-induced gene silencing (HIGS) and genes editing of susceptibility genes by *e.g.*, CRISP-Cas9 are most promising, though no resistant varieties have been released based on NGT (*Collinge et al., 2021*). *Groza, Bowen & Kichefski (2004)* also tested different potato varieties and realized that there was different susceptibility level among varieties.

Seed-borne diseases are controlled principally through one of the two notions. (i) To ensure that pathogen free seeds are planted to anticipate the emergence of disease. (ii) To seek ways that would eliminate disease pathogens from infected or contaminated seeds before the latter are planted. Pathogen exclusion by detection and elimination of infested seed lots is required for the management of serious diseases like bacterial wilt (*Etebu & Nwauzoma, 2017*). On the other hand, it was recommended that seed tubers should not be used as a seed more than two cycles (*Kassa, 2019*). In Ethiopia, different initiatives were involved to set-up community or farmer group-based seed production and attempts were made to establish and capacitate cooperatives to produce and supply quality seed potato for potato growers in different parts of the country. Unfortunately, most of the cooperatives assessed in southern Ethiopia were not functioning for production and marketing quality seed as expected or some were performing unlikely very weak (*Tadesse et al., 2020*; *Ayano, 2019*). Among many factors that influence the function of the cooperatives identified were tension between perspective rules, collective action and individual interest of the members in that cooperative or farmer groups (*Tafesse et al., 2018*; *Tadesse et al., 2020*).

## Improved agronomic practices

From the time when agriculture began, various improved agronomic practices were progressed by generations of farmers and expertise. Scholars developed new crop management systems, hence the use of non-chemical methods for preventing plant disease is a priority (*Meynard, Doré & Lucas, 2003*). To overcome the spread of potato bacterial wilt, planting clean seed tubers followed by continuous field inspection has a significant role minimize the spread of *R. solanacearum* (*Forbes et al., 2020*; *Tessema et al., 2020*). The field sanitation and crop management practices have dual benefits; to avoid the survival of the pathogen on crop fields and thereby limiting its dissemination to other disease-free areas through different disease spreading mechanisms (*Priou et al., 1999*; *PHAU, 2018*). Developing innovative, eco-system friendly and efficient agronomic methods are essential to overcome the major production constraints such as pests and diseases (*Karim & Hossain, 2018*; *Tadele, 2017*). *Nyawade et al. (2016)* pointed out incorporating suitable indeterminate legume cover crops such as Dolichs lablab in potato cropping systems enabled to minimize soil and nutrient losses due to erosion. This in turn has effect on soil acidity, which has positive association with bacterial wilt prevalence in potato fields (*Tafesse et al., 2021*). The study by *Mwaniki et al. (2017)* indicated that rotations involving spring onion with the locally grown cereals such as barley and wheat could be utilized in curbing bacterial wilt in Kenya.

Such innovative agronomic practices can help crops compute effectively against pests and reduce excessive chemical use while controlling pests as well as improving yield and quality of the produce (*Stephen & Nora, 2002*). Agronomic activities like topsoil amendment with 5% to 10% farmyard manure suppressed bacterial wilt severity and pathogen survival in soil as well as improved tomato yield in Ethiopia

(*Yadessa, van Bruggen & Ocho, 2010*). Other agronomic practices such as crop rotation and mixing crop varieties had also significant effect on bacterial wilt incidence in potato (*Kassa, 2019*; *Wang et al., 2021*). In the study conducted by *Lu et al. (2016)* with two different biochar made from peanut shell and wheat straw were added to *R. solanacearum* infested soil and the results showed that both treatments significantly reduced disease severity by 28.6% and 65.7% respectively in tomato fields. Soil management activities like biochar application in potato field also improved tuber yield and reduced red ant infestation in the soil (*Upadhyay et al., 2020*).

## Biological control

Biological control in the abstemious sense that denotes the use of antagonistic microbes to combat the pathogen and as such represents a potentially sustainable approach. Biological control entails the use of natural enemies such as, predators, parasitoids, and pathogens to manage pest problems. Over recent years, biological control has progressed to the stage where a number of products are on the market with four major recognized mode of actions such as; competition, antibiosis, hyper parasitism, and induced resistance (*Collinge et al., 2021*). Nevertheless, a weakness of biological control lies in its vulnerability to environmental factors. Sometimes, biological control is a "stand alone" method and does not have to be used in combination with other methods. This is especially true for effective natural enemies being used against pest in uncultivated areas, aquatic weeds, rangeland weeds, or arthropod pests of ornamental plants or in forests, all of these being ecologically stable habitats usually requiring lower levels of management.

Natural enemies have key roles in pest management programs worldwide. Using natural enemies in pest management requires an understanding of their basic biology, how they impact pest population growth, and how the environment and management system affect natural enemy dynamics and performance (*O'Neil & Obrycki, 2010*). When pesticides must be applied to combat one pest, they should be applied so that they do not kill natural enemies controlling other pests in the same system (*Hajek, 2012*).

Biological control has long been considered as a possible alternative to strategies for pest control, however its effect and level of use worldwide remain modest and inconsistent (*Gurr & You, 2016*). Out of 77 bacterial strains isolated from six soil rhizospheres samples and six vegetal material samples of healthy potato, four strains (E7, E13, S25, and P7) shown high antagonistic activity against *R. solanacearum* with soil zones of inhibition from 23 to 40 mm with various degrees of disease incidence and biocontrol efficacy in Madagascar (*Rado et al., 2015*). *Kurabachew, Assefa & Hiskias (2007)* characterized and evaluated 50 *Pseudomonas fluorescence* from Ethiopian isolates as bio-control agent against potato bacterial wilt caused by *R.solanacearum* and only three of the isolates showed inhibition against the growth of the pathogen. The study confirmed that bacterization of tubers with the above three isolates significantly reduced the incidence of *R. solanacearum* by 59.83% compared to the pathogen-inoculated control and suggests the importance of the isolates as plant growth-promoting rhizobacteria. On the other study by *Aguk et al. (2018)* confirmed that BCAs were effective against bacterial wilt even in the susceptible cultivar evaluated under controlled condition.

## Chemical control

Chemical control is the use of pesticides to significantly reduce the impact of pests on desirable plants by killing, suppressing growth, inhibiting biological functions, and/or disrupting behavior patterns. Chemical control of BW is ineffective, the use of healthy seeds and pathogen-free soil and water, as well as crop rotation, are the principal means of control (*Alvarez, Biosca & Lopez, 2010*). The use of chemical pesticides in potato is increasing in developing countries due to farmers intensifying their production beyond the crops traditional range (*FAO, 2009*). Despite, a clear increase in pesticide use, crop losses have not significantly decreased during the last 4 decades (*Oerke, 2006*). The increasing dose of pesticide application by smallholder farmers in Ethiopia had an adverse impact on the environment and human health (*Ayana & Fufa, 2019*). Combating pests with the intensive use of pesticides can harm the environment and pose a serious threat to the health of producers and consumers as well; hence regular monitoring of potato farms for pests and the broader agro-ecosystem is the basis for eco-friendly plant protection and pest management (*Lutaladio et al., 2009*).

## Integrated disease management (IDM)

The goal of integrated disease and pest management (IPM) is to adapt sustainable and more resilient crop production systems independent on pesticide application. IPM requires a good knowledge and skill of individual potato production systems; identifying pest species, knowing their biology and symptoms of infestation to decide and undertake appropriate technique on their integrated management (*Kroschel et al., 2020*; *FAO, 2017*; *Uwamahoro et al., 2018*). Although many of proved pest control technologies provide significant economic benefits when employed in a suitable manner, IPM is a key component of sustainable agriculture. Moreover, the advantage of IMP for farmers in developing countries has been clear for many, though its implementation is relatively limited (*Asfaw, 2019*).

Complexity of *R. solanacearum* such as variations based on host plant, cultivar, climate, soil type, cropping practice, pathogen strain and others hampered the success of searching effective disease control method and the endorsements made by different scholars embraces the implementation of integrated disease management approach for better-off crop production (*Mihovilovich et al., 2017*; *Karim & Hossain, 2018*; *Kassa, 2019*).

The source of inoculum could be infected potatoes (seed tubers, harvest leftovers and volunteer plants) or infected soils or both (*Abdurahman et al., 2017*). Moreover, irrigation water, other plant debris or other field tools and animal movements could be the spreading agents of the pathogen as well (*FAO, 2017*; *PHAU, 2018*; *EPPO, 2018*; *Williamson, Hudelson & Allen, 2002*). To control and eradicate bacterial wilt, it is essential to implement all the possible disease control options simultaneously with intensively integrated approach (*Thomas-Sharma et al., 2016*). The use of healthy seed potato and planting in disease-free soil is the main components although other factors must under consideration while implementing integrated disease management techniques (*Priou et al., 1999*). Hence, interdisciplinary seed system development and integrated disease management approaches could also be possible options to overcome such complex

problems (*Sperling et al., 2013*; *Thomas-Sharma et al., 2016*). Knowing the disease cycle is one important aspect to implement appropriate and environmentally friendly disease control methods. Strong extension systems and inclusive community participation on knowledge sharing among the neighbors and the whole society could give a chance for farmers to reduce the spread of bacterial wilt from place to place (*Ayano, 2019*; *Damtew et al., 2018*; *Tafesse et al., 2018*). Moreover, considering gender on research on pests and diseases is very crucial as it facilitates development of more efficient approach thereby increase the adoption of crop protection technologies and practices by women and men farmers according to their knowledge, role, and capacities (*Kawarazuka et al., 2020*).

As no conventional disease control method has been found effective alone for the bacterial wilt management, implementing comprehensive and combined disease control techniques are noble options for sophisticated pathogens like *R. solanacearum* (*Karim & Hossain, 2018*; *Uwamahoro et al., 2018*). Rotating individual fields away from crops within the same family is critical and can help minimizing crop specific disease and non-mobile insect pests that persist in the soil or overwinter in the field or field borders (*Seaman, 2016*; *Kassa, 2016*). In response to the increasing plant pest problems, community-based plant clinics were introduced to Ethiopia in 2013 and played significant role in meeting farmer demand for plant health advice and in bridging the gabs of local diagnostic capacity. The new approach is guided by IPM practices and recommends safe, economical and practical pest management options for smallholder farmers in Ethiopia (*Efa & Feleke, 2019*).

## STATUS OF BW IN ETHIOPIA

Since the first bacterial wilt occurrence story in the country in 1956 (*Stewart, 1956*), scholars made tremendous efforts for the last 6 decades to know the status of the disease and its prevalence (Table 1). The status of bacterial wilt was studied from simple pathogen survey and strain identification to molecular characterization to the sequevar level. Currently, the disease is threatening potato production in Ethiopia (*Gorfu, Woldegiorgis & Kassa, 2013*; Abdulwahab et al., 2017). Different scholars made efforts from awareness creation to recommendations to the farming community for pathogen management (*Ayano, 2019*; *Kassa, 2016*, *Tafesse et al., 2018*). Moreover, recent studies were focused on understanding and managing bacterial wilt, analysis of its monitoring systems by seed cooperatives with combining and innovative systems approach, collective action and a system thinking perspectives to magnify the socio-economic implications of *R. solanacearum* for smallholder farmers in Ethiopia (*Damtew et al., 2018*; *Tafesse et al., 2018*, *2020*). In addition to the efforts made by the government of Ethiopia, NGOs like CIP and Vita (an Irish NGO) contributed a lot to manage potato bacterial wilt in the country (*Sharma et al., 2018*; *Ayano, 2019*; *Tadesse et al., 2020*). Nevertheless, research results from 261 randomly surveyed potato farmers in Gumer, Doyogena, and Wolmera districts revealed that they had very limited knowledge about potato bacterial wilt and 60% of the farmers did not know spreading mechanisms of the pathogen (*Tafesse et al., 2018*). Similarly, *Ayano (2019)*, who conducted an assessment on farmers' extension approach for the control of potato bacterial wilt in Chencha district of the southern Ethiopia, reported

considerable knowledge gap existence among farmers. However, almost all farmers in the study area were willing to engage in collective action to combat bacterial wilt. The study also suggested that both male and female headed farmers need access to information, skills, and tools to improve potato yields with more emphasis on extension service to female and resource poor households (*Ayano, 2019*).

Hence, most potato farmers have limited knowledge of potato bacterial wilt and a comprehensive and community-based mobilization on disease control actions could be a better approach to mitigate the current potato production threat of Ethiopia and other nations worldwide (*Tafesse et al., 2018*). On the contrary, about 60% of potato farmers in the central highlands of Ethiopia were able to identify symptoms, causative agents and spreading mechanisms, as well as possible management actions of potato bacterial wilt, late blight and viruses after subsequent extension support, field demonstration and experimentation conducted at farmers field school level (*Kassa, 2016*).

In most developing world, inaccessibility of good quality seed and lack of farmer knowledge on proper agronomic practices were among the constraints that endanger the sustainability of the potato crop (*Priou et al., 1999*). Furthermore, the status of plant quarantine and pest management service in Ethiopia is not well organized and the status of major economic pests have not been routinely surveyed, monitored, and their seasonal situation have not been properly documented or communicated due to the loose linkage existence among the federal and regional bureau of agriculture (*Beyene & Salato, 2019*). Seed regulatory institutions in the country also lacks autonomy and role clarity between the federal and regional seed health regulatory bodies and currently the federal ministry of agriculture and regional bureaus of agriculture oversee regulation of the seed system and other inputs (*Ministry of Agriculture and Ethiopian Agricultural Transformation Agency, 2016*). Owing such weak government structural organization, regular pest management task has been left for farmers to handle every aspect by their own indigenous knowledge (*Beyene & Salato, 2019*).

## PERSPECTIVES

Although agriculture is the backbone of Ethiopian economy, agricultural production and productivity remained very low due to several factors contributing to the backwardness of the sector (*Damtew et al., 2018*). Among these, there is weak phytosanitary service in the country that has several challenges and gaps with limited capacity, insufficient facility and skilled manpower which could not be able to protect crop losses due to aggressive pests (*Beyene & Salato, 2019*).

Preliminary evidence has been presented which suggests that the combined implementation of current policy, including systematic plant health testing and continuous survey programs and exploration of research results is having a positive effect on the control of potato bacterial wilt in the country. The goal of eradicating the pathogen from Ethiopia will require a sustained and non-complacent effort in which policy makers; industry, research institutions and other stakeholders must all play an important role.

Access to extension to farmers is limited, but it has the power to mitigate disease spread in the farming community and about the fate of the disease to food security (*Ayano, 2019*). On the other hand, environmentally friendly alternatives for pesticides such as bio-organic fertilizers have been developed or better control of soil-borne diseases and used as a prime example of next-generation sustainable agriculture (*Aguk et al., 2018*). Hence, designing and practicing integrated disease (IDM) strategies which are knowledge-intensive process of decision making that combines various approaches (biological, physical, cultural, and chemical, regular field monitoring and inspection) should be shared for reducing the level of bacterial wilt disease (*Seaman, 2016*).

## CONCLUSIONS

This review addresses the complicated nature and importance of the devastating potato bacterial wilt disease likely to influence food security. An alarming increase in the number of outbreaks of transboundary pests and diseases of plants and animals are threatening food security and have broad economic, social, and environmental impacts in the world. Diseases are complex problems for crop production with numerous technical and institutional features, involving multiple actors with diverse perceptions and understandings. Crop pests continues to incur substantial post-harvest losses on many important crops and caused 30% yield loss in horticultural crops alone and the yield loss increased up to 50% in Ethiopia for some individual crops. Potato bacterial wilt becomes the most production limiting factor in Ethiopia. Awareness creation on potato bacterial wilt and integrated disease management approach could bring fundamental impact to the farming community mostly to smallholder farmers in developing countries.

Additional recommendations for future research that arises from this review are:

1. Currently, potato bacterial wilt in Ethiopian highlands and central highlands is under national pandemic peak that controls all most all seed potato domain areas of the country with more than 90% disease prevalence and spreading to previously uninfected localities. This issue is time sensitive and needs prompt attention. Hence, the government authorities, research institutions and all stakeholders should take a comprehensive disease management action to rescue the vibrant potato sector of Ethiopia.

2. Efforts to avoid zero tolerance diseases especially in seed potato should be maximized.

3. Disease preventive rules on potential symptomless carriers like latently infected seed tubers should be well implemented and the research on BW should not be simple field survey or laboratory work, rather they should exercise both fundamental and applied research on the topic.

4. Implementing appropriate seed legislation and quarantine rules could also requires due attention in Ethiopia to safeguard the potato crop for food security.

5. Community-based interventions and monitoring approaches for potato bacterial wilt also could be a solution to mitigate the disease and would enable us to win the disease eradication game.

## ACKNOWLEDGEMENTS

The authors thank the Holetta Agricultural Research Centre for providing necessary facilities.

### Funding

The authors received no funding for this work.

### Competing Interests

The authors declare that they have no competing interests.

### Author Contributions

- Gebrehanna Lemma Tessema conceived and designed the experiments, performed the experiments, prepared figures and/or tables, authored or reviewed drafts of the article, and approved the final draft.
- Hussen Ebrahim Seid performed the experiments, authored or reviewed drafts of the article, and approved the final draft.

### Data Availability

This is a literature review and does not have raw data.

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
