# Peer review of "Potato bacterial wilt in Ethiopia: history, current status, and future perspectives"

_PeerJ, doi:10.7717/peerj.14661_

## Round 0.1 · original submission · Major Revisions

Dear Authors,

Thank you for submitting your work to PeerJ. I have received the reviews and would like to inform you that your work has been received with moderate revision requests. I agree with the reviewers that the manuscript needs to undergo an English check and rigorous review by a proficient English speaker or Technical writer.

Reviewer 2 suggested reducing the text; however, I leave this to the author's discretion so that the author can decide what the readers can read and understand from their work. But I highly appreciate Reviewer 2's valuable time to put considerable effort to improve the work as you can see from the annotated file. Also Reviewer 3 suggested to present your thoughts in a tabular form.

Hence the reason stays here to consider this work to undergo major revision. However, I believe the authors can easily address all the issues. Please address these changes and resubmit.

Once again Thank you very much for choosing PeerJ.

Dr. Nagendran Tharmalingam
Academic Editor- PeerJ

·

Basic reporting

The manuscript submitted by Gebrehanna et.al entitled “Potato bacterial wilt in Ethiopia: history, current status, and future perspectives” talk about the potential risk affecting the production of potatoes, a staple food in Ethiopia due to bacterial wilt is also known as brown rot. The authors decided to make this study due to the increasing spread of this potato wilt to regions that were previously less affected by this disease. By doing this the authors aim to boost the production of potatoes in the region which is under deep pressure due to food security. The authors make use of the available database for their study to help accomplish the bacterial wilt by giving a comprehensive understanding of its past, present, and future perspective and ultimately giving some information for the techniques which can be used for targeting the bacterial wilt.

The manuscript is well written with a brief introduction about the subject and the reason why it is important to study the disease. However there are a few points about the manuscript which are as follows
The authors need to correct the typos where ever possible

The authors need to correct lines 54 to 55 “to an ever-expanding and Africaís second largest population which”

The authors should correct line 235 “as potato bacterial wilt has no territory limits and it could be”

The authors should correct lines 382 to 383 “and leads to additional 2 billion people will leave in developing countries”

The authors should correct lines 390 to 391 “have been quite differ”

The authors used the acronym NGM correctly or it has to be NGT at line 444

The authors can make a box highlighting the key takeaways of the manuscript and also add the outstanding questions which need to be addressed by the field.

Overall the work done by Gebrehanna et.al is commendable.

Experimental design

No comment.

Validity of the findings

No comment

Additional comments

No comment.

Reviewer 2 ·

Basic reporting

The manuscript is contains relevant information regarding to the status of potato bacterial wilt in Ethiopia, with emphasis on the search for combating the disease. Control strategies cannot be indicated if knowledge is not well established. Therefore, I encourage the authors to improve the text to make it more suitable for publication. Detailed suggestions for improvement are in the attached PDF file. A

Experimental design

The manuscript is contains relevant information regarding to the status of potato bacterial wilt in Ethiopia, with emphasis on the search for combating the disease. Control strategies cannot be indicated if knowledge is not well established. Therefore, I encourage the authors to improve the text to make it more suitable for publication. Detailed suggestions for improvement are in the attached PDF file. Among the suggestions, I indicate the major ones below:
1. Keep the text focused on the subject, which is potato bacterial wilt. Sometimes, the text is too general, indicating other diseases and other pathogens. A general review on bacterial wilt would be too extensive due to an extensive host range of the pathogen, and therefore not appropriate to meet the objective indicated in the title.
2. Do a careful language review.
3. Do a careful review also on technical terms, such as italicizing latin names of the pathogens and hosts, whenever appropriate. It includes the references at the endo of the manuscript.
4. Review the figures and tables, keeping focus on potato and Ralstonia solanacearum.
5. The figure on the worldwide distribution of R. solanacearum has alraedy been published. Omit it.

Validity of the findings

Since it is a review article, no comments here.

Additional comments

Please check the attached PDF file for detailed instructions or comments.

Annotated reviews are not available for download in order to protect the identity of reviewers who chose to remain anonymous.

Reviewer 3 ·

Basic reporting

The review article by Gebrehanna et al nicely summarizes the history, current management, and future perspectives of Bacterial Wilt disease. Addition of pictorial demonstrations (For example life cycle of the pathogen) would make it easy for the reader to go through the article.
The article which focuses on Ethiopia, also describes the world scenario briefly, there they missed to discuss many important details related to other part of the world like India and African countries.
One of the important aspect the article clearly missed is the scientific details associated with the causing organism, its size, shape, nature etc.
The authors have provided a large detail of numbers which would be best represented as tables, so adding more tabular data will increase the visibility of the article.
Few of the abbreviations like NGT were not described in the text
Line 383 leave will be live

Experimental design

No comment

Validity of the findings

No comment

---

## Round 0.2 · Minor Revisions

Dear Authors,

Thank you for submitting the revised version of your manuscript with suggested modifications. We are happy to receive the revised work, and the reviewers have gone through the work. I must thank you for the work you have done with significant modifications. However, reviewer 3 requested a few minor modifications in terms of reducing the text in the Introduction and occurrence section. In addition, reviewer 3 suggested a pictorial or diagrammatic representation of part of the work that will reduce the text significantly. Hence, I request you to take this minor revision and submit your revised version.

Thank you for submitting your work with us, and I am looking forward to seeing the revised work.

Regards,
Dr. Nagendran Tharmalingam
Handling Editor.

·

Basic reporting

No comments

Experimental design

No comments

Validity of the findings

No comments

Additional comments

No comments

Reviewer 3 ·

Basic reporting

The authors have reduced the text, however, I feel there s significant scope to reduce it and make it more visible. Since the authors have made a point to stick to reducing the length, I think they can reduce other parts of the write-up like Introduction and Occurrence, and either add a separate topic or merge the brief life cycle of the pathogen. Also, it's always advisable to make pictorial representations than word heavy paragraphs.

Experimental design

NA

Validity of the findings

NA

---

## Round 0.3 · accepted · Accept

Dear Authors,

I am happy to inform you that the reviewers acknowledged the manuscript "Potato bacterial wilt in Ethiopia: history, current status, and future perspectives" can be published in PeerJ, and we are glad to publish your MS with us. The production team will contact you for further type-set-related questions. Best wishes.

Regards,
Dr. Nagendran Tharmalingam
Academic Editor

Reviewer 3 ·

Basic reporting

NO comments

Experimental design

N/A

Validity of the findings

n/A